# Computational design of high-performance ligand for enantioselective Markovnikov hydroboration of aliphatic terminal alkenes

Hiroaki Iwamoto[1], Tsuneo Imamoto[2,3] & Hajime Ito [1]

Finding optimal chiral ligands for transition-metal-catalyzed asymmetric reactions using trial-and-error methods is often time-consuming and costly, even if the details of the reaction mechanism are already known. Although modern computational analyses allow the prediction of the stereoselectivity, there are only very few examples for the attempted design of chiral ligands using a computational approach for the improvement of the stereoselectivity. Herein, we report a systematic method for the design of chiral ligands for the enantioselective Markovnikov hydroboration of aliphatic terminal alkenes based on a computational and experimental evaluation sequence. We developed a three-hindered-quadrant P-chirogenic bisphosphine ligand that was designed in accordance with the design guidelines derived from this method, which allowed the Markovnikov hydroboration to proceed with high enantioselectivity (up to 99% ee).

[1] Division of Applied Chemistry, Graduate School of Engineering, Hokkaido University, Sapporo, Hokkaido 060-8628, Japan. [2] Organic R&D Department, Nippon Chemical Industrial Co., Ltd., Kameido, Koto-Ku, Tokyo 136-8515, Japan. [3] Department of Chemistry, Graduate School of Science, Chiba University, Yayoi-cho, Inage-ku, Chiba 263-8522, Japan. Correspondence and requests for materials should be addressed to H.I. (email: hajito@eng.hokudai.ac.jp)

Asymmetric catalysis is one of the most sophisticated ways to induce controlled molecular transformations[1]. Since the humble beginnings of this research area in the 1960s, the discovery and optimization of chiral catalysts for asymmetric reactions have strongly depended on experimental trial-and-error methods, even though these conventional methods are usually time-consuming and/or expensive. Although recent high-throughput screening techniques have improved the discovery efficiency[2–4], the preparation of catalyst libraries for complicated catalyst structures and their synthesis remains usually highly laborious. Moreover, especially when very high levels of enantioselectivity are required, it is unlikely that a better catalyst can be found by chance through screening of a chiral catalyst library or structural modifications of the state-of-the-art catalyst without any rational guidance for the improvement of the selectivity. Recent developments of density functional theory (DFT) calculations have enabled scientists to acquire information on possible reaction paths, including intermediates and transition-state structures, and these modern methods, which enclose long-range dispersion correlations, have reached a high level of accuracy with respect to the prediction of the stereoselectivity in organic reactions[5–10]. Meanwhile, multidimensional analysis methods based on steric parameters and properties of organic compounds for asymmetric reactions have been developed[11, 12]. However, studies, wherein the computational results were used effectively for the rational design of enantioselective catalysts remain scarce[13–16]. Our basic motivation was therefore to design asymmetric catalysts by using a combination of computational and experimental evaluations (Fig. 1a).

As the target reaction, we focused on regio- and enantioselective transformations of aliphatic terminal alkenes (α-olefins), which are feedstock chemicals obtained from petrochemicals. Their asymmetric functionalization with high selectivity (>95% ee) remains challenging, due to the low steric demand and the small electronic differentiation between the terminal and internal carbon atoms[17]. As the target of our catalyst-design project, we selected the asymmetric hydroboration reactions of aliphatic terminal alkenes using a copper(I)-catalyst (Fig. 1b). Catalyzed and uncatalyzed hydroborations of terminal alkenes proceed under anti-Markovnikov regioselectivity[18, 19]. In contrast, the development of hydroboration reactions of terminal alkenes that proceed under Markovnikov regioselectivity remains a long-standing challenge. In 1989, Hayashi and co-workers realized the first enantioselective Markovnikov hydroboration of styrene

substrates with a chiral rhodium catalyst[20–25], and the thus obtained enantioenriched secondary alkylboronates have become versatile intermediates in organic chemistry[26, 27]. Achieving high Markovnikov selectivity for aliphatic terminal alkenes, under catalyzed or uncatalyzed conditions, has been more difficult. In 2016, our group first reported that an achiral copper(I)-catalyst that contains bulky 1,2-bis(diarylphosphino)benzene ligands provides racemic secondary alkylboronates with Markovnikov regioselectivity[28–34]. During the course of our study, an enantioselective Markovnikov hydroboration of aliphatic terminal alkenes has been reported by Aggarwal and co-workers, who used a chiral rhodium(III) catalyst and an asymmetric diboron reagent[35]. Very recently, Shi and co-workers have reported a copper(I)-catalyzed reaction using chiral NHC ligands[36]. Even though these developments are indisputably remarkable, the levels of enantioselectivity and/or regioselectivity are not as high as required for some applications: for example, the transformation of 4-phenyl-1-butene resulted in 80% ee, branch/linear = 98:2[35] and 96% ee, branch/linear = 80:20[36], respectively.

To establish the computational-study-assisted ligand design sequence, an ideal phosphine-ligand-preparation system is required that should be characterized by synthetic and structural modularity with regard to the substituents on the phosphorus atoms in order to implement the design guidelines obtained from the computational study, and a rigid core scaffold to avoid drastic structural changes upon changing the substituents on the phosphorus atoms. In accordance with these requirements, we chose P-chirogenic 2,3-bis(phosphino)quinoxaline ligands, as these are easily obtained from the stepwise coupling of a chiral phosphine module (CP) and an achiral phosphine module (AP) into 2,3-dichloroquinoxaline (Fig. 1c)[37–40].

Given the target reaction and ligand preparation procedure, we based the three-step ligand-design cycle on a combination of a computational analysis and an experimental evaluation: the first step is the experimental evaluation of a ligand for the regio- and enantioselectivity in the borylation of an aliphatic terminal alkene, the second step is the DFT calculation on the borylation with the ligand, followed by confirming the validity of the calculations by comparing the experimental and calculated selectivities, which delivers design guidelines in the form of a quadrant-by-quadrant structural analysis of the transition states, the third step is the synthesis of ligands via the modular coupling of CP, AP, and the 2,3-dichloroquinoxaline core. Through these iterative cycles, we finally identified a chiral ligand that shows

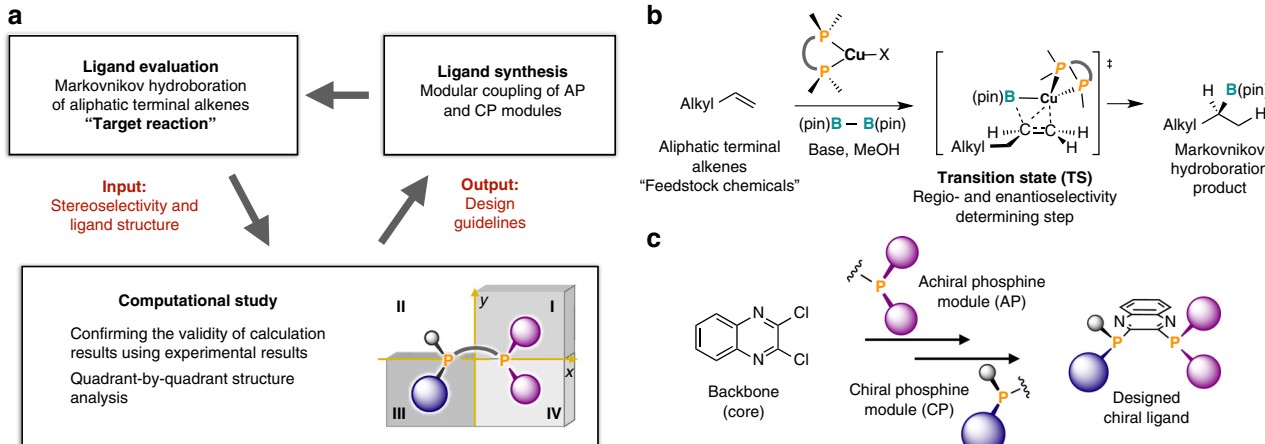

**Fig. 1** Ligand-design strategy based on a combination of computational and experimental evaluations. **a** Iterative chiral-ligand-optimization cycle. **b** Copper (I)-catalyzed enantioselective Markovnikov hydroboration of aliphatic terminal alkenes. [(pin) = pinacolato] **c** Modular synthesis of quinoxaline-based P-chirogenic bisphosphine ligands

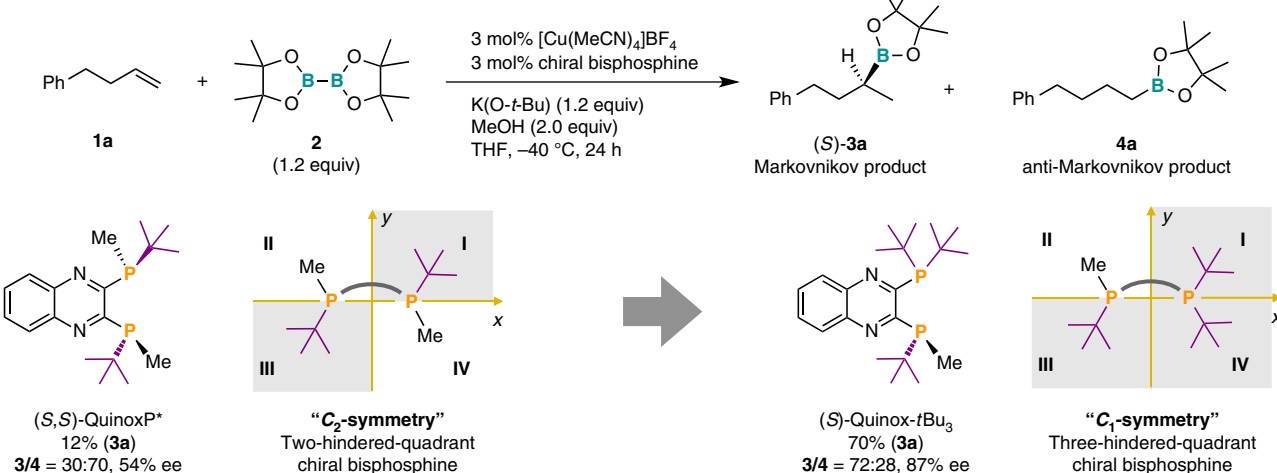

**Fig. 2** The first ligand evaluation step. Yields (%) of **3a** and regioselectivities (**3/4**) were determined by the gas chromatography (GC) analysis using the crude material. Enantioselectivities (% ee) were determined by the high-pressure liquid chromatography (HPLC) analysis with a chiral column

high levels of enantio- and regioselectivity (99% ee, br/l = 92:8) in the enantioselective Markovnikov borylation of aliphatic terminal alkenes.

## Results

**Experimental evaluation of the chiral ligand.** Initially, we screened commercially available chiral ligands (see Supplementary Table 1) in the reaction of **1a** using a copper(I) precursor, stoichiometric amounts of diboron, a base, and methanol as the proton source. We found that even though the reaction with a $C_2$-symmetric P-chirogenic bisphosphine ligand, i.e., (S,S)-QuinoxP*[38], delivered the undesired linear product **4a** as the major product, it showed at least moderate enantioselectivity [12% (**3a**), **3/4** = 30:70, 54% ee (S)-**3a**] (Fig. 2). In order to improve the selectivity, we subsequently modified the steric demand of the catalyst. The use of the more hindered $C_1$-symmetric three-hindered-quadrant P-chirogenic bisphosphine ligand (S)-3H-QuinoxP* [(S)-Quinox-$tBu_3$][40] inverted the regioselectivity and improved the enantioselectivity to give the branched product as the major product [70% (**3a**), **3/4** = 72:28, 87% ee (S)-**3a**]. This experimental catalyst investigation indicated that a three-hindered-quadrant chiral bisphosphine ligand should be most suitable for the regio- and enantioselective recognition of aliphatic terminal alkenes.

**Computational study based on (S)-Quinox-$tBu_3$.** To understand the observed stereoselectivity of the first generation of the chiral ligand (S)-Quinox-$tBu_3$, we carried out a computational study using a method that includes the dispersion correlation [ωB97XD/SDD for the Cu atom, 6-311G(d,p) for all other atoms in gas phase][41]. Initially, we compared the experimental and computational results of the regio- and enantioselectivity to check the validity of the calculation method and the basis set. As a model reaction, we used the reaction between (pin)B–Cu/(S)-Quinox-$tBu_3$ and 1-butene. In the case of an addition reaction between a $C_1$-symmetric borylcopper(I) complex and a mono-substituted terminal alkene, it is necessary to consider at least eight routes to understand the stereoselectivity (Fig. 3a): two reaction pathways ($A_{branch}$ and $D_{branch}$) to the major enantiomer (S)-**3**, which was obtained with (S)-Quinox-$tBu_3$, while another two reaction pathways ($B_{branch}$ and $C_{branch}$) lead to the minor enantiomer (R)-**3**, and the other four paths ($A_{linear}$, $B_{linear}$, $C_{linear}$, and $D_{linear}$) afford the linear product **4**. The activation energy

values of the transition states obtained from the DFT calculations (Fig. 3b) indicated that (i) for the major enantiomer (S)-**3**, path $D_{branch}$ is the most favorable reaction route [ +13.1 (0.0) kcal mol$^{-1}$], (ii) for the linear product **4**, path $A_{linear}$ is the most favorable route [+13.7 (+0.6) kcal mol$^{-1}$], and (iii) for the production of the minor enantiomer (R)-**3**, path $B_{branch}$ is a major pathway [+14.5 (+1.4) kcal mol$^{-1}$]. The activation barriers of the other pathways [TS-$A_{branch}$: +15.0 (+1.9) kcal mol$^{-1}$; TS-$C_{branch}$:+16.0 (+2.9) kcal mol$^{-1}$; TS-$B_{linear}$:+16.0 (+2.9) kcal mol$^{-1}$; TS-$C_{linear}$: +15.6 (+2.5) kcal mol$^{-1}$; TS-$D_{linear}$:+16.8 (+3.7) kcal mol$^{-1}$] should be too high to contribute to the product selectivity. The enantio- and regioselectivity estimated based on the calculated activation energy values for the borylcupration of the carbon–carbon double bond were closely matched with the experimental result of **1a** (predicted values: 83% ee, **3/4** = 76:24; experimental values: 87% ee, **3/4** = 72:28). This excellent match between computational and experimental results convinced us about the accuracy of the calculations on the copper(I)-catalyzed Markovnikov hydroboration of aliphatic terminal alkenes.

We then tried to extract important structural factors that affect the regio- and enantioselectivity by an analysis of steric congestion for each quadrant, i.e., a quadrant-by-quadrant analysis of the following three important transition states for the stereoisomers: TS-$D_{branch}$ for (S)-**3** (66.5%) (Fig. 3c), TS-$B_{branch}$ for (R)-**3** (6.1%) (Fig. 3d), and TS-$A_{linear}$ for **4** (22.6%) (Fig. 3e). The most favorable transition state structure, TS-$D_{branch}$, revealed that the pinacolato boryl moiety [B(pin)], which inclines in direction of the vacant quadrant **II**, renders the rigid chiral space suitable for recognition of the prochiral carbon–carbon double bond by avoiding the steric repulsion between the methyl group in the boryl moiety and the alkyl chain of 1-butene (Fig. 3c) (see Supplementary Fig. 8). In contrast, TS-$B_{branch}$, which affords the minor enantiomer, should be destabilized by the steric congestion between the alkyl chain of 1-butene and the methyl group in the B(pin) moiety, evident from the short distance between H4 (1-butene) and H6 [B(pin)] (Fig. 3d). Strain in this structure should also arise from the steric interaction between a methyl group in the B(pin) moiety and the tert-butyl group of the ligand in quadrant **I**. Additionally, the tert-butyl group in quadrant **III** and the alkyl chain of 1-butene are relatively close in TS-$B_{branch}$. The difference of the activation energy values between TS-$D_{branch}$ and TS-$A_{linear}$ is relatively small (+0.6 kcal mol$^{-1}$). The steric congestion between the tert-butyl groups of the ligand in quadrants **III** and **IV** with the B(pin) moiety in TS-$A_{linear}$ distorts the transition state (Fig. 3e). A structural comparison of TS-$D_{branch}$

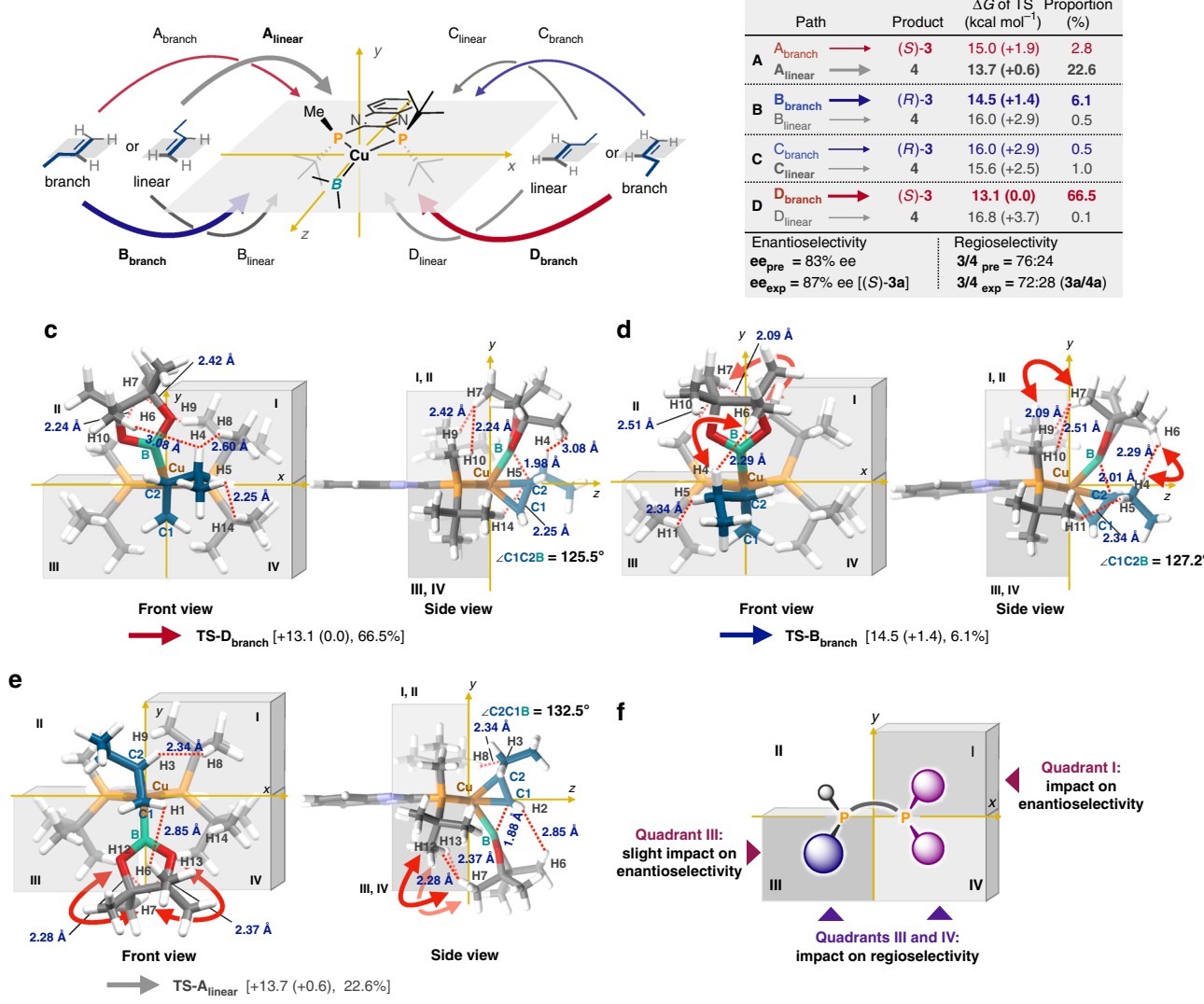

**Fig. 3** The first computational study step with (S)-Quinox-*t*Bu₃. **a** Eight possible reaction routes to products **3** and **4**. **b** DFT-calculated activation energy values. **c** Structural analysis of TS-D_branch. **d** Structural analysis of TS-B_branch. **e** Structural analysis of TS-A_linear. **f** Design guidelines from a quadrant-by-quadrant structural analysis

and TS-B_branch suggests that the steric demand in quadrants **I** and **III** plays a major and a minor role for the enantioselectivity, respectively. In contrast, according to a structural comparison of TS-D_branch and TS-A_linear, the steric congestion in quadrants **III** and **IV** significantly affects the regioselectivity (Fig. 3f).

**Synthesis and evaluation of a series of Quinox-type ligands.** To achieve higher enantioselectivity and regioselectivity than (S)-Quinox-*t*Bu₃, we synthesized several Quinox-type three-hindered-quadrant chiral bisphosphine ligands via a stepwise modular synthesis using an electrophilic quinoxaline core with the corresponding phosphine modules (**AP**s and **CP**s) on the basis of the design guidelines derived from the computational study (Fig. 3f) as the ligand synthesis and evaluation steps (Fig. 4). The chiral bisphosphine ligand (S)-Quinox-Ad*t*Bu₂, which consists of a chiral adamantyl methylphosphino [(S)-Ad: **CP2**] and di-*tert*-butyl phosphino modules (*t*Bu₂: **AP1**), is bulkier than (S)-Quinox-*t*Bu₃ in quadrant **III**, and exhibited slightly higher regioselectivity and enantioselectivity than (S)-Quinox-*t*Bu₃ (69%, 91% ee, **3/4** = 82:18). This improvement of the selectivities upon structurally modifying (S)-Quinox-*t*Bu₃ into (S)-Quinox-Ad*t*Bu₂ is thus consistent with

the structural analysis (Fig. 3f). The use of the chiral ligand (S)-Quinox-*t*BuAd₂, which bears chiral *tert*-butyl methylphosphino modules [(S)-*t*Bu: **CP1**] and sterically demanding adamantyl groups (Ad₂: **AP2**) in quadrants **I** and **IV**, dramatically improved the enantioselectivity (66%, 97% ee, **3/4** = 82:18), while the regioselectivity remained similar to that of (S)-Quinox-Ad*t*Bu₂, which is again consistent with the structural analysis (Fig. 3f). (S)-Quinox-Ad₃, which was synthesized by combining phosphine modules **CP2** and **AP2**, afforded excellent enantioselectivity and good regioselectivity (85%, 97% ee, **3/4** = 86:14). These selectivity improvements are clearly consistent with the DFT-derived guidelines for the design of chiral bisphosphine ligands (Fig. 3f). Nevertheless, the regioselectivity could still be improved.

**Computational study with (S)-Quinox-Ad₃.** To gather more detailed information in order to further improve the enantioselectivity and the regioselectivity of the second generation of the chiral bisphosphine ligand (S)-Quinox-Ad₃, we moved on to a computational study step, which was concerned with DFT calculations on the borylcupration step of the terminal alkene in the presence of (S)-Quinox-Ad₃ (Fig. 5). According to the activation energy values

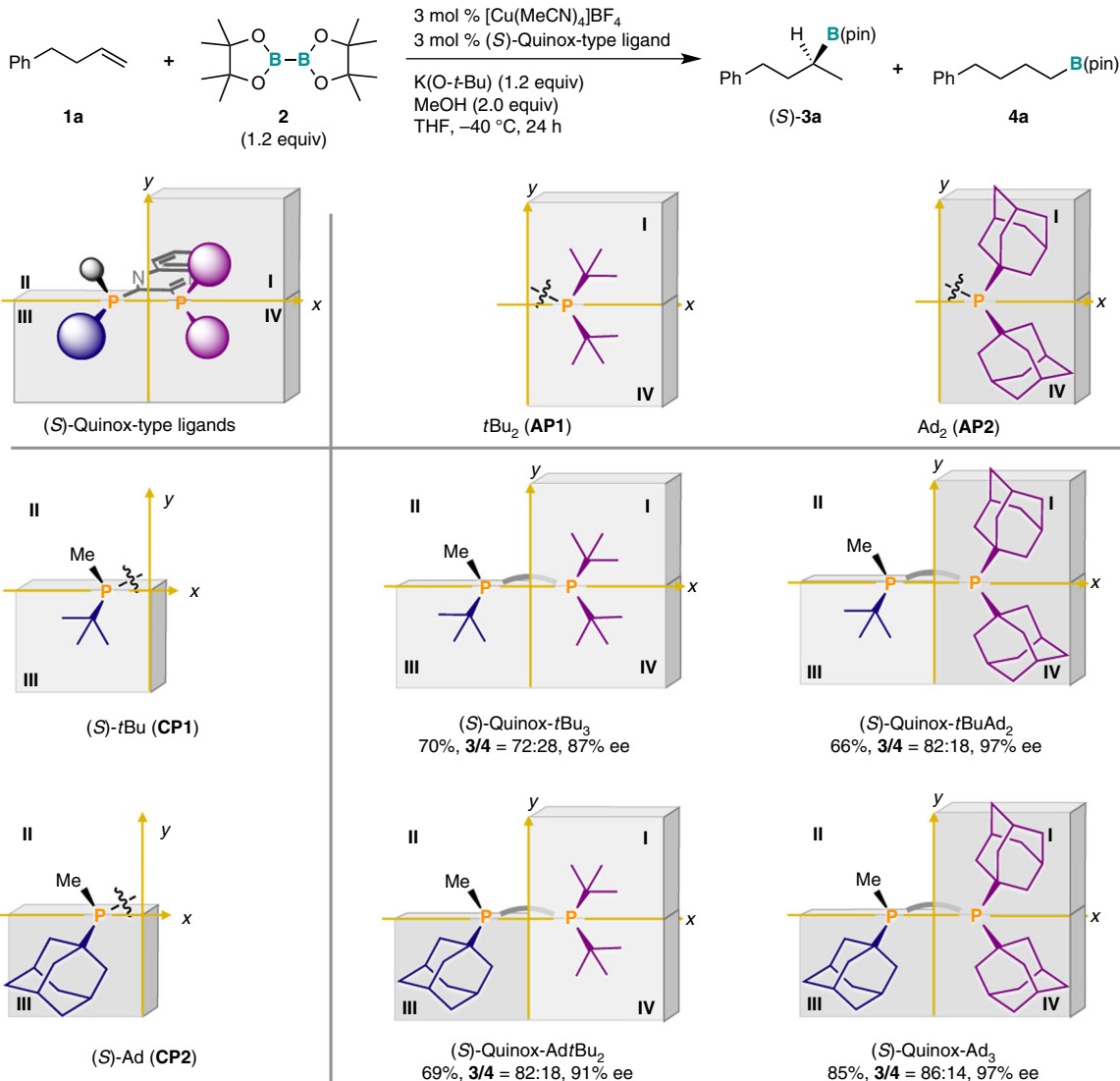

**Fig. 4** The first ligand synthesis step with the design guideline and the second ligand evaluation step. Yields (%) of **3a** and regioselectivities (**3/4**) were determined by the gas chromatography (GC) analysis using the crude material. Enantioselectivities (% ee) were determined by the high-pressure liquid chromatography (HPLC) analysis with a chiral column

of the transition states, the predicted enantioselectivity and regioselectivity were adequately matched with the experimental results (predicted values: 99.6% ee, **3/4** = 90:10; experimental values: 97% ee, **3/4** = 86:14, Fig. 5b). The transition state TS-D$'_{branch}$, which provides the major enantiomer (S)-**3**, is the most favorable reaction pathway (+12.3 kcal mol$^{-1}$), while TS-B$'_{branch}$, which provides the minor enantiomer (R)-**3** was effectively destabilized [+17.4 (+5.1) kcal mol$^{-1}$] by the bulkier adamantyl substituents in quadrants **I** and **IV** as compared to (S)-Quinox-tBu$_3$ [TS-B$_{branch}$ = + 14.5 (+1.4) kcal mol$^{-1}$] (Fig. 3d). Similarly, TS-A$'_{linear}$ was also slightly destabilized [+13.8 (+1.5) kcal mol$^{-1}$] by changing the tert-butyl groups in (S)-Quinox-tBu$_3$ to an adamantyl group in (S)-Quinox-Ad$_3$. While TS-C$_{branch}$ [+16.0 (+2.9) kcal mol$^{-1}$] for (S)-Quinox-tBu$_3$ scarcely contributed to the product ratio, TS-C$'_{branch}$ [+15.9 (+3.6) kcal mol$^{-1}$] became the lowest pathway for the minor enantiomer (R)-**3**.

Subsequently, we carried out the quadrant-by-quadrant structural analysis of two important unfavorable pathways, i.e., TS-C$'_{branch}$ and TS-A$'_{linear}$. In TS-C$'_{branch}$, steric congestion was observed between the B(pin) and adamantyl moieties in quadrants **III** and **IV** (Fig. 5c). The steric interaction in quadrant

**III** should exert a more pronounced influence on the activation barrier of TS-C$'_{branch}$ than that in quadrant **IV**, as the B(pin) moiety is inclined in direction of quadrant **III** to avoid the steric interaction with the alkyl chain of 1-butene and the ligand in quadrant **IV**. In TS-A$'_{linear}$ (Fig. 5d), the adamantyl groups of (S)-Quinox-Ad$_3$ in quadrants **III** and **IV** should interact more effectively with the B(pin) group relative to the tert-butyl group of (S)-Quinox-tBu$_3$, which is reflected in the larger energy difference between TS-D$'_{branch}$ and TS-A$'_{linear}$ [(S)-Quinox-Ad$_3$:+1.5 kcal mol$^{-1}$] compared to that between TS-D$_{branch}$ and TS-A$_{linear}$ [(S)-Quinox-tBu$_3$:+0.6 kcal mol$^{-1}$]. It should also be noted that the adamantyl groups in (S)-Quinox-Ad$_3$ are sterically more demanding in the periphery of the reaction center (quadrants **III** and **IV**), which is more effective in controlling the steric repulsion between the ligand and the B(pin) moiety than the tert-butyl groups in the first-generation ligand (S)-Quinox-tBu$_3$.

**Discovery of the high-performance chiral ligand**. From the quadrant-by-quadrant structural analysis of the second-generation chiral ligand (S)-Quinox-Ad$_3$ (Fig. 5c and 5d), we

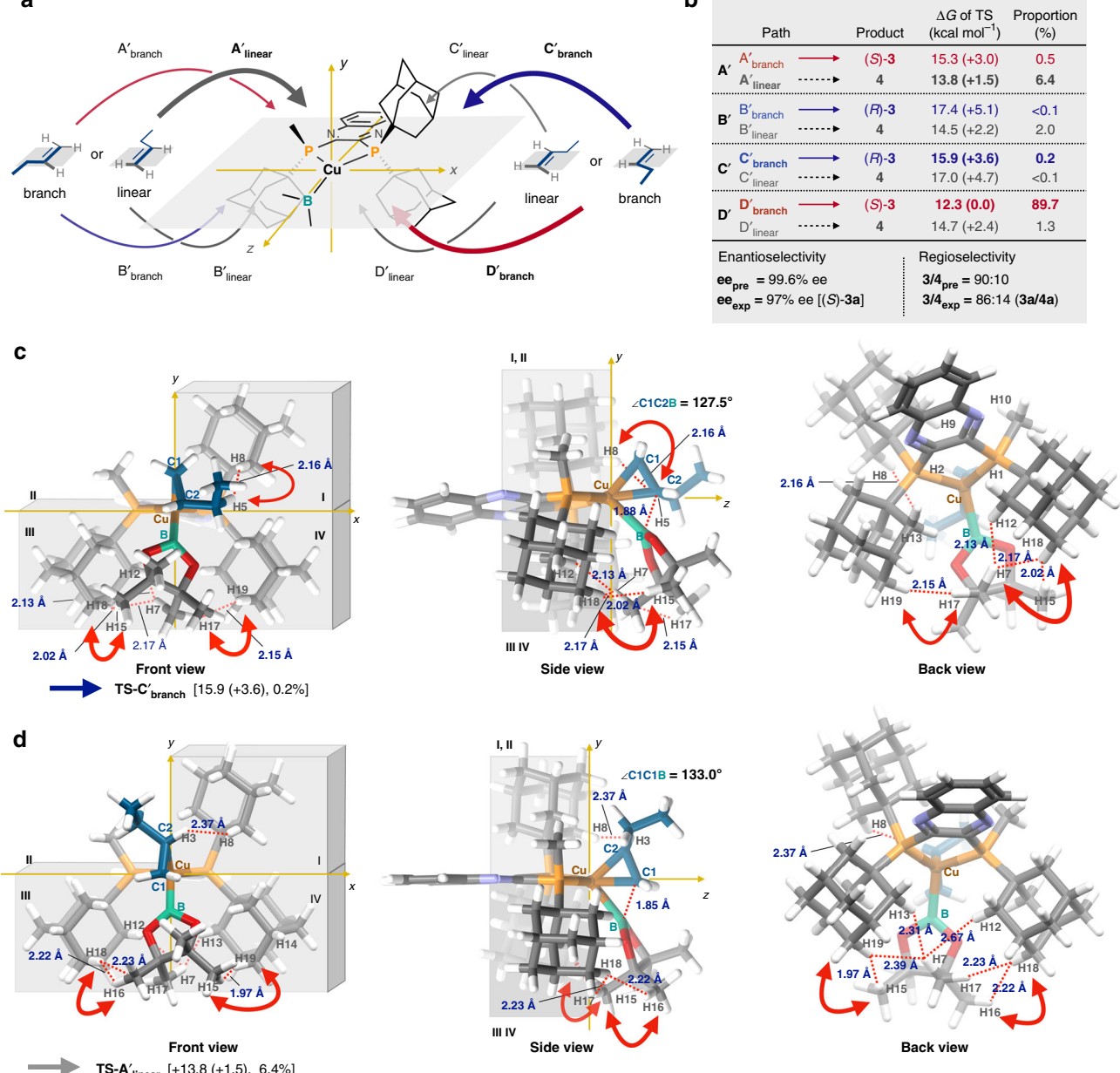

**Fig. 5** The second computational study step with (*S*)-Quinox-Ad₃. **a** Eight possible reaction pathways. **b** Activation energy values for the eight possible reaction routes together with the predicted enantioselectivity and regioselectivity values. **c** Structural analysis for TS-C′branch. **d** Structural analysis for TS-A′linear

obtained the design guidelines for the improvement of the stereoselectivity (Fig. 6). We anticipated that the introduction of a substituent that is sterically more demanding in the periphery of the reaction center (quadrant **III**) than the adamantyl group should destabilize TS-C′branch, while the introduction of a sterically highly demanding substituent in the periphery should destabilize TS-A′linear. Therefore, we focused on the *tert*-octyl group, which is bulkier than the 1-adamantyl group according to the steric parameters in a quantitative structure-activity relationship (QSAR)[42–44]. The Charton value (*v*), which corresponds to Taft's steric parameter correlated with the van der Waals radii, of the *tert*-octyl group is larger than that of the 1-adamantyl group (*tert*-butyl: 1.24; 1-adamantyl: 1.33; *tert*-octyl: 1.74). Furthermore, the Sterimol parameter B5 of the *tert*-octyl group, which reflects the long-range steric bulk, is larger than that of the 1-adamantyl group (*tert*-butyl: B5 = 3.17; 1-adamantyl: B5 = 3.49;

*tert*-octyl: B5 = 4.54). Based on these considerations, we identified, an unknown ligand, (*S*)-Quinox-*t*OctAd₂, which contains a *tert*-octyl group in quadrant **III** as the optimal chiral ligand[45]. We then synthesized (*S*)-Quinox-*t*OctAd₂ and found that it exhibited almost perfect enantioselectivity and high regioselectivity to afford the desired Markovnikov hydroboration product in high yield (92% for **3a**, 99% ee, **3/4** = 92:8). This enantioselectivity was higher than the previously reported values [Rh: 80% ee[35]; Cu: 96% ee[36]].

**Substrate scope.** With (*S*)-Quinox-*t*OctAd₂ as the optimized chiral bisphosphine ligand for the enantioselective Markovnikov hydroboration of terminal aliphatic alkenes in hand, we examined the substrate scope (Table 1). The reactions of aliphatic terminal alkenes (**1a–e**), including linear α-olefins (LAOs) such as 1-octene

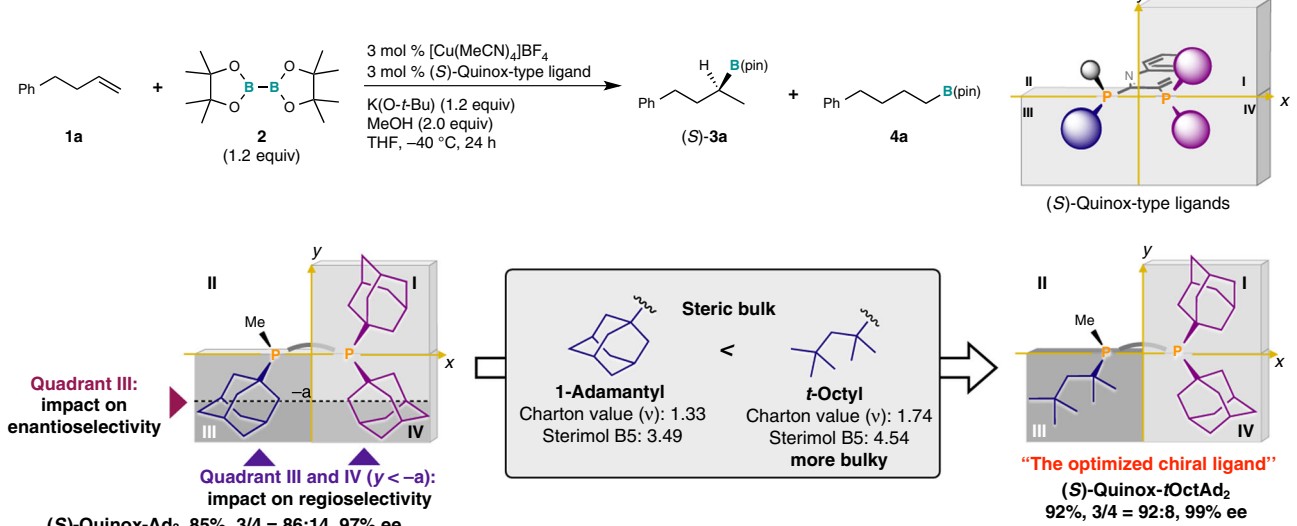

**Fig. 6** The second ligand synthesis step with the design guideline and the third ligand evaluation step. Yields (%) of **3a** and regioselectivities (**3/4**) were determined by the gas chromatography (GC) analysis using the crude material. Enantioselectivities (% ee) were determined by the high-pressure liquid chromatography (HPLC) analysis with a chiral column

**Table 1 Substrate scope of the enantioselective Markovnikov hydroboration of aliphatic terminal alkenes.[a]**

| | | | | |
|---|---|---|---|---|
| (S)-**3a**[b]: 94% 3/4 = 92:8, 99% ee | (S)-**3b**[b]: 76% 3/4 = 88:12, 98% ee | (S)-**3c**[b]: 61% 3/4 = 89:11, 99% ee 71% (5.0 mmol scale) 3/4 = 89:11, 98% ee | (S)-**3d**[b]: 57% 3/4 = 88:12, 97% ee | (S)-**3e**: 88% 3/4 = 85:15, 99% ee |
| (S)-**3f**: 60% 3/4 = 93:7, 98% ee | (S)-**3g**: 89% 3/4 = 86:14, 98% ee | (S)-**3h**[b]: 73% 3/4 = 84:16, 99% ee | (S)-**3i**[c]: 93% 3/4 = 87:13, 97% ee | (S)-**3j**: 91% 3/4 = 90:10, 98% ee |
| (S)-**3k**: 85% 3/4 = 90:10, 96% ee | (S)-**3l**: 80% 3/4 = 89:11, 96% ee | (S)-**3m**: 71% 3/4 = 88:12, 95% ee | (S)-**3n**: 54% 3/4 = 90:10, 95% ee | (S)-**3o**: 67% 3/4 = 83:17, 95% ee |

[a]Conditions: **1** (0.5 mmol), [Cu(MeCN)$_4$]BF$_4$ (0.025 mmol), (S)-Quinox-tOctAd$_2$ (0.025 mmol), **2** (0.6 mmol) and K(O-t-Bu) (0.6 mmol) in THF (1.0 mL) at −40 °C. Isolated yield. Regioselectivities (**3/4**) were determined by the gas chromatography (GC) analysis using the crude material. Enantioselectivities (% ee) were determined by the high-pressure liquid chromatography (HPLC) analysis with a chiral column. [b]3 mol % catalyst loading. [c]At −10 °C.

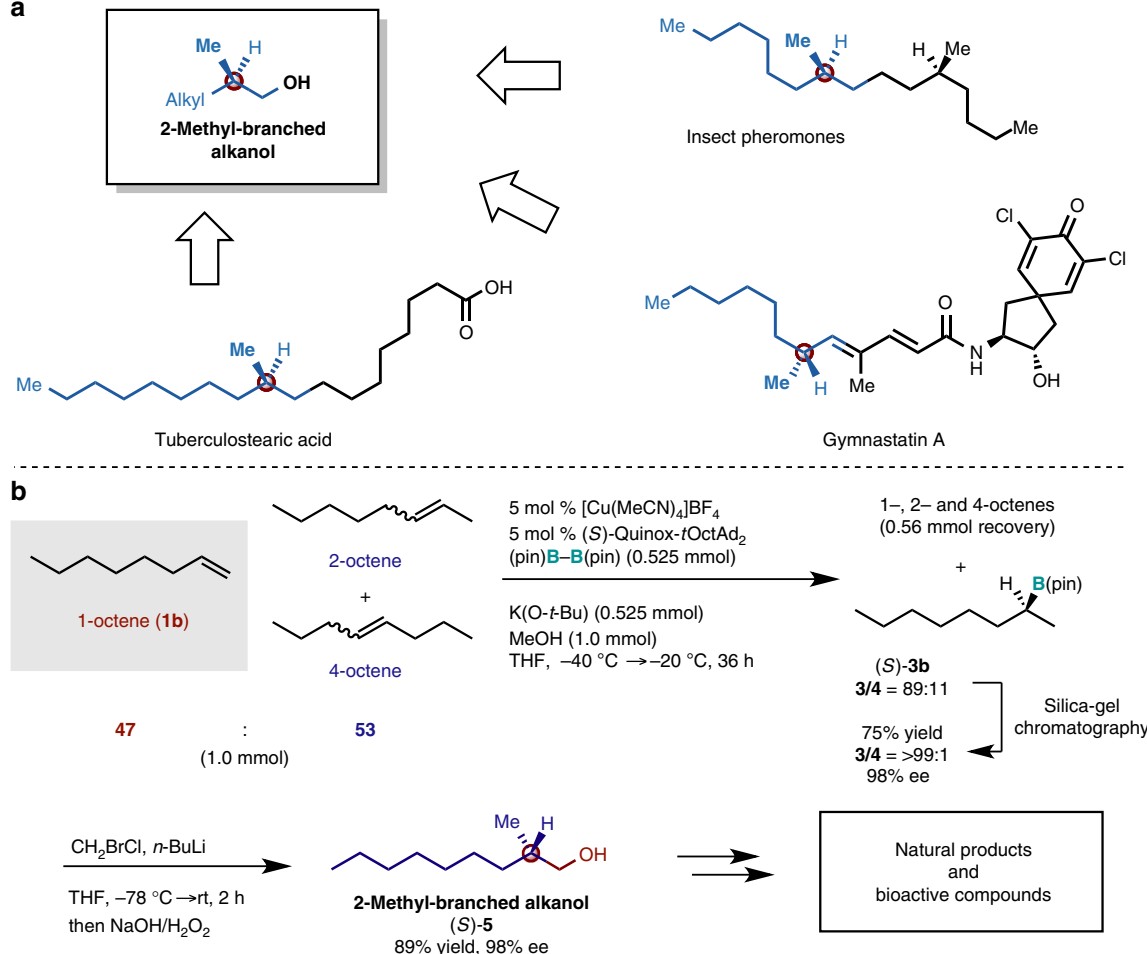

**Fig. 7** Synthetic applications of the Markovnikov hydroboration products. **a** Chiral 2-methyl alkanols as versatile intermediates for natural products or bioactive compounds bearing chiral methyl-branched moieties. **b** Synthesis of 2-methyl-alkanol (S)-**5** via the enantioselective Markovnikov hydroboration of a mixture of octenes

(**1b**), 1-decene (**1c**), and 1-dodecene (**1d**) afforded the corresponding secondary alkylboronates (S)-**3a–e** with high regioselectivity and excellent enantioselectivity. This is the first example for a catalyst that promotes the Markovnikov borylation of LAOs in high selectivity [Rh catalyst: (S)-**3c**, 80% ee[35]; Cu catalyst: (S)-**3d**, 91% ee[36]]. The reactivity and stereoselectivity were not affected by large-scale reaction conditions. We also conducted reactions with terminal alkenes that contained functional groups. The reaction of allylsilane **1f** afforded (S)-**3f** with high regioselectivity and enantioselectivity. Secondary alkylboronates containing halide moieties were obtained in high yield with moderate regioselectivity and excellent enantioselectivity, whereas the reactions of substrates bearing oxygen atoms as functional groups or a bulky alkyl substituent resulted in slightly lower, but still sufficiently high enantioselectivity ( > 95% ee).

**Synthetic applications.** 2-Methyl-branched alkanols are versatile intermediates for the synthesis of insect pheromones, chiral saturated carboxylic acids found in glycolipids, and other natural products (Fig. 7a)[46–49]. Conventional synthetic procedures for methyl-branched alkyl chains often require stoichiometric amounts of optically active starting materials. The potential utility of our enantioselective Markovnikov hydroboration was demonstrated on the synthesis of the enantioenriched 2-methyl-alkanol (S)-**5** from an octene mixture that included 1-octene (**1b**), cis- and trans-2-octene, as well as cis- and trans-4-octene, which

can be considered as a model substrate for a petrochemical feedstock obtained from a cracking process, where the isomers are difficult to separate (Fig. 7b). This catalytic system only reacted with the terminal alkene to provide the borylation product (S)-**3b** with good regioselectivity (**3/4** = 89:11), while the internal alkenes were recovered almost quantitatively. Other borylation products that could have been potentially generated from internal alkenes were not detected. The minor linear product **4b** was removed from the crude material by column chromatography to give pure (S)-**3b**. The subsequent homologation of (S)-**3b** was accomplished by a treatment with an oxidant, which furnished chiral 2-methyl-branched alkanol (S)-**5** in good yield with excellent enantioselectivity (89% over two steps, 98% ee).

## Discussion

We have developed a rational design strategy for asymmetric catalysts based on an experimental and computational evaluation sequence, which afforded the high-performance chiral bisphosphine ligand (S)-Quinox-tOctAd$_2$ for the enantioselective Markovnikov hydroboration of aliphatic terminal alkenes. The key to success in this approach is the modularity of the quinoxaline-based ligand structure and the introduction of appropriate phosphine modules in accordance with the design guidelines obtained from the computational investigation (quadrants **I** and **III**: enantioselectivity; quadrants **III** and **IV**: regioselectivity). The optimized bisphosphine ligand (S)-Quinox-tOctAd$_2$ exhibited a broad substrate scope for

terminal aliphatic alkenes and good functional-group tolerance to furnish a variety of secondary chiral alkylboronates with high regioselectivity and excellent enantioselectivity (up to 99% ee). Ultimately, this strategy should help to pave the way to the computer-assisted design of asymmetric catalysts.

## Methods

**General Procedure for the Enantioselective Hydroboration.** [Cu(MeCN)$_4$]BF$_4$ (4.7 mg, 0.015 mmol), bis(pinacolato)diboron (**2**) (152.4 mg, 0.60 mmol) and chiral ligand (0.015 mmol) were placed in an oven-dried reaction vial. After the vial was sealed with a screw cap containing a Teflon$^{TM}$-coated rubber septum, the vial was connected to a vacuum/nitrogen manifold through a needle. It was evacuated and then backfilled with nitrogen. This cycle was repeated three times. Dry THF (0.40 mL) and K(O-$t$-Bu)/THF (1.00 M, 0.60 mL, 0.60 mmol) were added in the vial through the rubber septum using a syringe. After stirring for 30 min at −40 °C, **1** (0.50 mmol) and methanol (0.0404 mL, 1.0 mmol) were added to the mixture at −40 °C. After the reaction was complete, the reaction mixture was passed through a short silica-gel column (Φ: 10 mm, height of the silica-gel column: 30 mm) eluting with Et$_2$O. The regioselectivity of **3** was determined by GC analysis. The crude material was purified by flash column chromatography (SiO$_2$, Et$_2$O/hexane, typically 0:100–3:97) to give the corresponding alkylboronate (S)-**3**. The enantioselectivity was determined by HPLC analysis with a chiral column after the stereospecific derivatization of the boryl group.

**Data availability.** The X-ray crystallographic coordinate for the structure of [(S)-Quinox-Ad$_3$] reported in this research article has been deposited at the Cambridge Crystallographic Data Centre (CCDC) under deposition number CCDC 1817859 [https://www.ccdc.cam.ac.uk/]. For full characterization data including NMR spectra of new compounds and experimental details, see Supplemental Information.

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

## Acknowledgements

This work was financially supported by JSPS KAKENHI grant JP15H03804 and 18H03907. H. Iwamoto would like to thank the JSPS for their support in form of a scholarship (grant number 16J0141006). We would like to thank Dr. Tomohiro Seki, Mr. Mingoo Jin, Mr. Koh Kobayashi, and Mr. Kentaro Kashiyama for carrying out the X-ray analyses. We also thank Prof. Satoshi Maeda for the valuable discussions.

## Author contributions

H. Iwamoto carried out the chemical experiments and the computational study. T. Imamoto instructed the synthetic method of the chiral phosphine ligands and supplied the phosphine intermediates for the phosphine ligands. H. Ito supervised the project. All authors discussed the results and commented on the manuscript.

## Additional information

**Competing interests:** The authors declare no competing interests.

