## [Peer Review File · Nature Communications]

REVIEWERS' COMMENTS:

Reviewer #1 (Remarks to the Author):

This paper describes a very impressive study on the design of chiral ligands for the enantioselective Markovnikov hydroboration of aliphatic terminal alkenes. Through a combined computational and experimental evaluation sequence Ito has developed a three-hindered-quadrant P-chirogenic bisphosphine ligand which resulted in Markovnikov hydroboration to proceed with high enantioselectivity (up to 99% ee) and regioselectivity. The interesting chiral ligands could have further applications in other processes. Interestingly, the substrate scope is quite broad and since it is selective for terminal alkenes it means that mixtures can be used and high chemoselectivity is observed. The paper is well written and gives a good balanced overview of the area. Some of the computational diagrams are a little more difficult to read but I think that is the nature of the science. In Fig 2b, it seemed that the major pathway had the alkene adding on the face with the two tBu groups (lower face) whereas the upper face seems less hindered with Me and tBu groups. The authors should check this again. P8 significant figures for % seem too many.

Reviewer #2 (Remarks to the Author):

A beautifully rendered work, stunning in execution, worthy of publication, and particularly well executed. The approach isn't genuinely novel, though the end result is quite a selective synthetic reaction. Computational design, the rationale, the quadrant model, and the general procedures are all routine among the leaders in the field. Is this among the best in the field, yes. Is this groundbreaking, no. Having said that, there are very few in the literature with this caliber of complete package from all angles. A masterpiece.

Reviewer #3 (Remarks to the Author):

The manuscript by Ito et al is conducting the development of a three-hindered-quadrant P-chirogenic bisphosphine ligand which allowed the Markovnikov hydroboration to proceed with high enantioselectivity (up to 99% ee). This is an interdisciplinary work since combine predictive theoretical studies and catalysis but they are not novel because other previous studies have used the same methodology to identify appropriate ligands in asymmetric hydrogenation. The application is also very specific, despite the fact that several substrates were hydroborated with success, but QuinoxP type ligands have been previously shown to be useful ligands in 1,4-hydroboration of α,β -unsaturated substrates, with Cu(I) catalysts (Synthesis of Functionalized Organoboron Compounds Through Copper(I) Catalysis, Spinger, Ed. Koji Kubota). In this work the conclusion are postulated from hypothesis that have precise understanding of the main problem but they can not be general for catalytic studies where other descriptors and terms modify the reaction outcome. The applicability of quinoxP ligands is already demonstrated (<https://doi.org/10.1002/anie.201701963>) therefore the manuscript lacks from novelty. It is not expected that this paper will make a big change on thinking in the field. This reviewer suggest that this manuscript can be submitted to a more specialized type of journal, such as ACS catalysis.

Reviewer(s)' Remarks to Author:

Reviewer: 1

Comments:

This paper describes a very impressive study on the design of chiral ligands for the enantioselective Markovnikov hydroboration of aliphatic terminal alkenes. Through a combined computational and experimental evaluation sequence Ito has developed a three-hindered-quadrant P-chirogenic bisphosphine ligand which resulted in Markovnikov hydroboration to proceed with high enantioselectivity (up to 99% ee) and regioselectivity. The interesting chiral ligands could have further applications in other processes. Interestingly, the substrate scope is quite broad and since it is selective for terminal alkenes it means that mixtures can be used and high chemoselectivity is observed. The paper is well written and gives a good balanced overview of the area. Some of the computational diagrams are a little more difficult to read but I think that is the nature of the science. In Fig 2b, it seemed that the major pathway had the alkene adding on the face with the two tBu groups (lower face) whereas the upper face seems less hindered with Me and tBu groups. The authors should check this again. P8 significant figures for % seem too many.

Comment 1-1:

- In Fig 2b, it seemed that the major pathway had the alkene adding on the face with the two tBu groups (lower face) whereas the upper face seems less hindered with Me and tBu groups. The authors should check this again.

Response 1-1:

We thank this the reviewer for the positive comment. Approaching the alkene from the upper face (quadrants I and II) seemed to be favored reaction pathway for the production of both enantiomers in the case of the use of three-hindered quadrant chiral ligand (*A_{branch}* and *C_{branch}*), however, the calculation results indicated that approaching the alkene from the lower face (quadrants III and IV) is favorable for the enantiomers (*D_{branch}* and *B_{branch}*). These unexpected results could be explained by the steric interaction between the B(pin) moiety and substituents of the ligand in their transition state structures. In the most favorable transition state (*TS-D_{branch}*), the pinacolato boryl moiety [B(pin)],

which inclines in direction of the vacant quadrant **II**, renders the rigid chiral space suitable for recognition of the prochiral carbon–carbon double bond by avoiding the steric repulsion between the methyl group in the boryl moiety and the alkyl chain of the alkene. Whereas, in the TS-*A*_{branch} and TS-*C*_{branch}, the relatively strong steric interaction between the B(pin) moiety and two *tert*-butyl groups would destabilize their transition state structures (see below).

Figure. Transition state structures by the DFT calculation for Markovnikov hydroboration products [ω B97XD/SDD for the Cu atom, 6-311G(d,p) for all other atoms in the gas phase].

Comment 1-2:

- P8 significant figures for % seem too many.

Response 1-2:

We corrected the significant figures. The significant figures of the proportions were defined the first decimal place.

Reviewer: 2

Comments:

A beautifully rendered work, stunning in execution, worthy of publication, and particularly well executed. The approach isn't genuinely novel, though the end result is quite a selective synthetic reaction. Computational design, the rationale, the quadrant model, and the general procedures are all routine among the leaders in the field. Is this among the best in the field, yes. Is this ground-breaking, no. Having said that, there are very few in the literature with this caliber of complete package from all angles. A masterpiece.

Response 2:

We really appreciate this positive comment.

Reviewer: 3

Comments:

The manuscript by Ito et al is conducting the development of a three-hindered-quadrant P-chirogenic bisphosphine ligand which allowed the Markovnikov hydroboration to proceed with high enantioselectivity (up to 99% ee). This is an interdisciplinary work since combine predictive theoretical studies and catalysis but they are not novel because other previous studies have used the same methodology to identify appropriate ligands in asymmetric hydrogenation. The application is also very specific, despite the fact that several substrates were hydroborated with success, but QuinoxP type ligands have been previously shown to be useful ligands in 1,4-hydroboration of α,β -unsaturated substrates, with Cu(I) catalysts (Synthesis of Functionalized Organoboron Compounds Through Copper(I) Catalysis, Spinger, Ed. Koji Kubota). In this work the conclusion are postulated from hypothesis that have precise understanding of the main problem but they can not be general for catalytic studies where other descriptors and terms modify the reaction outcome. The applicability of quinoxP ligands is already demonstrated (<https://doi.org/10.1002/anie.201701963>) therefore the manuscript lacks from novelty. It is not expected that this paper will make a big change on thinking in the field. This reviewer suggest that this manuscript can be submitted to a more specialized type of journal, such as ACS catalysis.

Response 3:

The main criticism raised by this reviewer is concerning the fact that QuinoxP* is well known and the copper-catalyzed borylation with QuinoxP* is not novel. This present paper is NOT for the classical QuinoxP* ligand BUT for new ligand design of chiral ligands possessing a quinoxaline backbone. The reviewer is confusing between previous studies on the classical QuinoxP* ligand and the present work. The classical QuinoxP* ligand bearing two *tert*-butyl groups with C_2 -symmetry have been proven the utility for various enantioselective reactions including copper-catalyzed borylations as the reviewer commented, while many of other possible Quinox-type ligands including various substituents have not been synthesized nor used for enantioselective reactions. In this work, we found and focused on synthesis of new Quinox-type ligands utilizing their synthetic modularity to reflect the design guidelines from the computational study. The

regio- and enantioselective transformations of aliphatic terminal alkenes with chiral catalysts have recognized as one of the most challenging issue in organic chemistry. Our catalyst design methods combined experimental and computational study with the Quinox-type ligands achieved the enantioselective recognition of aliphatic terminal alkenes with high accuracy (up to 99% ee). We believe that his method is applicable to other enantioselective reactions and improve the discovery efficiency of the optimal chiral catalyst.